# Stability of a Viable Non-Minimal Bounce

**Debottam Nandi** †

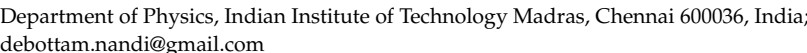

Department of Physics, Indian Institute of Technology Madras, Chennai 600036, India;
debottam.nandi@gmail.com
† Current address: Department of Physical Sciences, IISER Mohali, Manauli 140306, India.

**Abstract:** The main difficulties in constructing a viable early Universe bouncing model are: to bypass the observational and theoretical no-go theorem, to construct a stable non-singular bouncing phase, and perhaps, the major concern of it is to construct a stable attractor solution which can evade the Belinsky–Khalatnikov–Lifshitz (BKL) instability as well. In this article, in the homogeneous and isotropic background, we extensively study the stability analysis of the recently proposed viable non-minimal bouncing theory in the presence of an additional barotropic fluid and show that, the bouncing solution remains stable and can evade BKL instability for a wide range of the model parameter. We provide the expressions that explain the behavior of the Universe in the vicinity of the required fixed point i.e., the bouncing solution and compare our results with the minimal theory and show that ekpyrosis is the most stable solution in any scenario.

**Keywords:** bounce; inflation; conformal transformation; modified gravity; non-minimal coupling; stability; attractor

## 1. Introduction

In solving non-linear differential equations, initial conditions play a crucial role in determining the dynamical behavior of the system as different initial conditions may lead to different trajectories that may behave completely differently. Since gravity is highly non-linear in nature, solving early Universe evolution depends highly on the initial conditions as well. However, the already highly acceptable paradigm of the early Universe—the inflationary paradigm [1–13], perhaps succeeds based on the fact that, not only it is in line with the recent tighter observational constraints [14,15] but it also provides an attractor solution [16,17]. This implies that the early Universe inflationary solution does not depend on the initial conditions and because of this reason, even if the Universe contains additional matter(s), inflation makes them decay exponentially and the Universe solely dominates by the inflaton field responsible for inflationary evolution.

However, the greatest achievement comes from the fact that the inflationary solution can also evade Belinsky–Khalatnikov–Lifshitz (BKL) instability: the anisotropic energy density can grow faster than the energy density responsible for the evolution of the Universe [18] which can cause a highly unstable system. However, even with this much success, the inflationary paradigm also suffers many crucial difficulties. First, despite the ever-tightening observational constraints, there seem to exist many inflationary models that continue to remain consistent with the data [19–22], even leading to the concern whether inflation can be falsified at all [23]. Inflation being insensitive to initial conditions is still debatable. For instance, in Ref. [24], using non-perturbative simulations, authors showed that the inflationary expansion starts under very specific circumstances. In Ref. [25], authors pointed out the fact that small field potential fails to start the inflationary expansion under most initial conditions. It suffers from the trans-Planckian problem: the cosmological scales that we nowadays observe may correspond to length scales smaller than the Planck length at the onset of inflation, which in contradiction to the inherent assumption that the scales are classical even at the initial stage of inflation [26].

These issues lead to in search for alternatives to the inflationary paradigm and the most popular one is the classical non-singular bouncing scenario where, the Universe undergoes a phase of contraction until the scale factor reaches a minimum value, before it enters the expanding phase [27–32]. However, the problem with most bouncing models is that, it is extremely difficult to construct. While only known attractors are the ekpyrotic models [33], in fact, in Ref. [34], authors showed that the ekpyrotic contraction is a 'super-smoother', i.e., it is robust to a very wide range of initial conditions and avoid Kasner/mixmaster chaos and a similar statement could never be proven about any other primordial scenario, it fails to be in line with the observations. On the other hand, while matter bounce models may provide correct theoretical predictions consistent with the observations, the solution fails to be stable and thus, any tiny initial deviation can grow exponentially to cause an instability in the system. Another difficulty is, while constructing the contracting phase is rather easy to achieve, obtaining the non-singular bouncing phase is extremely difficult as it requires to violate the null energy condition: often called as the theoretical no-go theorem [35–44]. Most importantly, even if one may construct a model evading the instability and theoretical no-go theorem, perturbations suffer from many difficulties such as gradient instability, and these models fail to be in line with the observational constraints: a small tensor-to-scalar ratio ($r_{0.002} \lesssim 0.06$) and simultaneously, very small scalar non-Gaussianity parameter ($f_{\mathrm{NL}} \sim \mathcal{O}(1)$): referred to as the observational no-go theorem [45,46] (This has also been addressed in the past in the context of the so-called pre-big bang scenario based on the string cosmology equations, as reported for instance in a recent review in Ref. [47]). Apart from these main three issues, in general, bouncing models possess another, rather weak, difficulty as well: a natural exit mechanism from the bouncing phase to enter into the conventional reheating phase.

In solving these problems, it is soon realized that one needs to go beyond the canonical theories and consider non-minimal couplings, viz. the Horndeski theories or even beyond Horndeski theories [35,35–42,45,46,48–54]. However, it remains an open problem until in Ref. [55], we showed that a simple non-minimal coupling can solve all those issues.

In Refs. [56,57], we have shown that, the issue of stability can easily be resolved by using a simple non-minimal coupling. This has been achieved with the help of conformal transformation and we also have shown that, it may lead to viable tensor spectra as well [58]. Recently, we extended the work to construct the first viable non-minimal bouncing model which evades all the issues and is consistent with the constraints [55]. This model is constructed in such a way that the scalar sector of the action is conformal to that of the slow-roll inflationary model. Since, under conformal transformation, perturbations remain invariant and the constraints are obtained from the correlation functions of the scalar and tensor perturbations, we achieved to bypass observational no-go theorem. The choice of the coupling function also helps us to achieve the non-singular bouncing phase without any instability and therefore, it evades the theoretical no-go theorem and leads to the conventional reheating scenario. It has been pointed out that since the work is similar to works in Refs. [56,57], it can also resolve the issue of stability, mainly the BKL instability as well. However, the model contains a free model parameter $\alpha$ (which needs to be greater than zero to achieve the bouncing solution) and there is no bound on it. It is not clear how the system behaves in the presence of an additional matter.

At this moment, we need to stress that, when a model is studied for perturbations and is compared with the observations, it is assumed that, either the initial conditions are chosen with extreme measure, or the model solution is stable enough so that there is no need to fine-tune for choosing the initial conditions. Otherwise, even without the external matter, the perturbations themselves can cause instability as it also possesses a small amount of energy. This is the main reason why a stable solution is always preferred (one may even argue that only stable solutions are allowed). Therefore, even our model is conformal to inflationary solutions and obeys all relevant observational constraints, one needs to verify the stability of the model, especially in the presence of an external matter. This leads to the main motivation of this article.

In this work, we analyze the stability of the model in presence of an additional barotropic matter. The aim is to find the general attractor behavior of the bouncing solution that is consistent with the observations. There is mainly one free model parameter $\alpha$ (for minimal theory, $\alpha = -1$, since for this value, the coupling function becomes unity) along with the equation of state of the additional barotropic fluid $w_m$. In addition to that, since we consider conformal single field model that leads to slow-roll inflation, we also consider the potential slow-roll parameter in the minimal Einstein frame to be (nearly) constant (in case of exponential potential is strictly constant, as examined in Refs. [16,17]). With the help of these model parameters, we find the condition for stability for the bouncing solution and show that, to avoid the BKL instability leads to $\alpha < 2$. Since bouncing solution requires $\alpha > 0$, the bound becomes $0 < \alpha < 2$. To understand the behavior of the stability, we study the phase space in the vicinity of the fixed point corresponding to the bouncing solution and find that the system acts like the Universe contains three different types of matter and due to the attractor behavior, only the leading bouncing matter component remains intact, whereas, other two decay very fast. This helps us realize how the external barotropic fluid influences the system and how the attracting nature of the bouncing solution gets rid of the initial deviations.

A few words on our conventions and notations are in order at this stage of our discussion. In this work, we work with the natural units such that $\hbar = c = 1$, and we define the Planck mass to be $M_{\mathrm{Pl}} = (8\,\pi\,G)^{-1/2}$. We adopt the metric signature of $(-,+,+,+)$. We should mention that, while the Greek indices are contracted with the metric tensor $g_{\mu\nu}$, the Latin indices contracted with the Kronecker delta $\delta_{ij}$. Moreover, we shall denote the partial and the covariant derivatives as $\partial$ and $\nabla$. The overdots and overprimes, as usual, denote derivatives with respect to the cosmic time $t$ and the conformal time $\eta$ associated with the Friedmann-Lemaître-Robertson-Walker (FLRW) line-element, respectively. The sub(super)script '$I$' and $b$ denote the quantity in the minimal Einstein theory and non-minimal theory, respectively.

## 2. A Brief Introduction of the Bouncing Model

The non-minimal bouncing model is constructed in such a way that the scalar sector of the non-minimal action transforms conformally to that of a minimal canonical Einstein model that leads to slow-roll inflation. The slow-roll inflationary model can easily be constructed by using a single canonical scalar field $\phi$ minimally coupled to the gravity, i.e., the Einstein gravity as

$$\mathcal{S}_I = \frac{1}{2} \int \mathrm{d}^4\mathbf{x} \sqrt{-g_I} \Big[ M_{\mathrm{Pl}}^2 R^I - g_I^{\mu\nu} \partial_\mu \phi \partial_\nu \phi - 2\,V_I(\phi) \Big] + S_M^I(g_{\mu\nu}, \Psi_M). \tag{1}$$

$R^I$ is the Ricci scalar for the metric $g_{\mu\nu}^I$, and $V_I(\phi)$ is the potential responsible for the inflationary solution. The part of the action $S_M^I(g_{\mu\nu}, \Psi_M)$ indicates the presence of an additional fluid. As mentioned above, since inflation is an attractor, the evolution of the field does not depend on the additional fluid and is solely governed by the scalar sector of the system. Assuming the above theory is responsible for the slow-roll inflationary dynamics, the scale factor solution, during inflation, can approximately be written as a function of the scalar field $\phi$ as

$$a_I(\phi) \propto \exp\left( - \int^\phi \frac{\mathrm{d}\phi}{M_{\mathrm{Pl}}^2} \frac{V_I}{V_{I,\phi}} \right), \tag{2}$$

with $V_{I,\phi} \equiv \frac{\partial V_I}{\partial \phi}$. Now we shall construct a model which is conformal to the scalar part of the above inflationary action (1) in such a way that the new scale factor behaves as a bouncing solution. The transformation can be written as

$$g_{\mu\nu}^I = f^2(\phi)\, g_{\mu\nu}^b \quad \Rightarrow \quad a_I(\eta) = f(\phi)\, a_b(\eta). \tag{3}$$

$g^b_{\mu\nu}$ and $a_b(\eta)$ are the required bouncing metric and scale factor solutions, respectively, along with $f(\phi)$ being the coupling function. Using the above conformal transformation along with the action (1), we can construct the action responsible for such bouncing solution as

$$
\mathcal{S}_b = \frac{1}{2} \int \mathrm{d}^4\mathbf{x}\sqrt{-g_b}\left[ M^2_{\mathrm{Pl}} f^2(\phi)\, R^b - \omega(\phi)\, g^{\mu\nu}_b \partial_\mu\phi\partial_\nu\phi - 2\, V_b(\phi)\right] + S^b_M.
\tag{4}
$$

$\omega(\phi)$ and the potential $V_b(\phi)$ depend on the coupling function $f(\phi)$ and the inflationary potential $V_I(\phi)$ as

$$
\omega(\phi) = f^2(\phi)\left(1 - 6M^2_{\mathrm{Pl}}\frac{f_{,\phi}{}^2}{f^2}\right), \quad V_b(\phi) = f^4(\phi)\, V_I(\phi),
\tag{5}
$$

where $f_{,\phi} \equiv \frac{\partial f}{\partial \phi}$. $S^b_M$ is the external fluid present in the system in the non-minimal theory. Please note that, the above action resembles very similar to the gravi-dilaton string effective action, with the inclusion of non-perturbative string-coupling corrections [47]. Therefore, one can explain the existence of such kind of coupling functions in the domain of string theory. Note that, while the scalar part of the above action (4) is conformal to that of (1), these two actions in reality are not conformal due to the presence of the additional fluid. However, if the scalar field dominated solution in the non-minimal theory also becomes stable, similar to minimal theory, then, for a given inflationary model (1) with its solution (2), one can obtain the bouncing scale factor solution $a_b(\eta)$ in the non-minimal theory by setting $f(\phi)$ in a certain manner. For simplicity, if we desire the bouncing solution of the form

$$
a_b(\eta) \propto (-\eta)^\alpha \propto \exp\left(\alpha \int^\phi \frac{\mathrm{d}\phi}{M^2_{\mathrm{Pl}}}\frac{V_I}{V_{I,\phi}}\right), \quad \alpha > 0,
\tag{6}
$$

where $\eta$ is the comoving time, then by using Equations (2) and (3), one can obtain the required solution of the coupling function $f(\phi)$ as

$$
f(\phi) = f_0\, \exp\left(-\frac{(\alpha+1)}{M^2_{\mathrm{Pl}}}\int^\phi \mathrm{d}\phi\, \frac{V_I}{V_{I,\phi}}\right).
\tag{7}
$$

$f_0$ is normalized in such a way that at the minima of the potential, $f(\phi)$ becomes unity. The non-minimal theory (4) is the desired bouncing model: given an inflationary potential $V_I(\phi)$, we can completely determine the coupling function $f(\phi)$, which in turn, provides non-minimal theory (4) responsible for the bouncing scale factor solution (6). Note that, since Equation (2) is an approximated form, by using the above expressions (7) and (3), the obtained scale factor solution $a_b(\eta)$ is also not exact, but close to (6) (we will evaluate it later). We need to stress that the form of the scale factor (6) is valid only in the contracting phase, and during and after the bounce, it is extremely difficult to obtain an analytical solution. Therefore, the bouncing model, in our case, is also asymmetrical. The underlying assumption of the theory is that, either the bouncing solution is stable or we must choose a precise initial condition that leads to the bounce. Last of all, while $\alpha > 0$ provides a bouncing solution, there are actually no bound on $\alpha$ as it can take any value and negative values of it will lead to different Universe with different scale factor solution (6). For instance, $\alpha = -1$ brings us the conventional minimal Einstein scenario as, for this value, the coupling function becomes unity. In this work, we will concentrate only on the bouncing solutions where $\alpha > 0$ and obtain the condition for stability and the behavior of the system in the vicinity of the contracting solution, which we will explore in the next section.

Few things to note before we move in the next section. In case of scalar perturbation, in the Einstein frame (1), the perturbed action is

$$\int d\eta \, d\mathbf{x}^3 z_I^2 \left( \zeta_I'^2 - (\nabla \zeta_I)^2 \right), \quad z_I(\eta) \equiv \frac{a_I \phi'}{\mathcal{H}_I}, \tag{8}$$

where, $\zeta$ is the curvature perturbation and $\nabla$ is the divergence operator. In the non-minimal frame (4), the expression of the action for the scalar perturbation changes by replacing $z_I(\eta)$ with $z_b(\eta)$. However, $z_b(\eta)$ is simply the conformally transformed $z_I(\eta)$, i.e.,

$$z_b(\eta) = \frac{f(\phi) \, a_b(\eta) \, \phi'}{\left( \mathcal{H}_b + \phi' \frac{f_{,\phi}(\phi)}{f(\phi)} \right)} \tag{9}$$

This implies that, $z_b(\eta) = z_I(\eta)$ which leads to the action being identical in both frames and hence, $\zeta_I = \zeta_b$ at linear order. As the speed of sound is conformally invariant, $c_s$ is unity in both minimal inflationary as well as in non-minimal bouncing scenarios. Similarly, one can evaluate the perturbed interaction Hamiltonian (for detailed evaluation, see Refs. [59–63]) at any order and show that it also remains invariant in all conformal frames. This implies that the curvature perturbations at any order in both the frames are identical, which is not surprising, as we know, under conformal transformation, curvature perturbation remains invariant. The argument extends to tensor perturbation as well. Therefore, given a viable inflationary model which is consistent with the observations, in our bouncing model, there appear no divergences or instabilities (e.g., gradient instability) in the perturbations anywhere, even at the bounce and, in addition to that, the model also satisfies all observational constraints: thus evading the no-go theorem. The model leads to stable non-singular bounce as well as, shortly after the bounce, it leads to conventional reheating phase (for detailed information, see Figures 1 and 2, along with Ref. [55] as well as [64], to compare reheating in minimal theory with the same in non-minimal theory).

In short, for $\alpha > 0$, the model (4) leads to a stable, asymmetrical bouncing Universe, where the scale factor in the contracting region behaves as $a(\eta) \propto (-\eta)^\alpha$. If the slow-roll inflationary potential $V_I(\phi)$ satisfies the observational constraints (e.g., the Starobinsky potential), the corresponding bouncing model, irrespective of the value of $\alpha$, also satisfied all observational constraints, as the perturbations remain invariant under conformal transformation, thus evading the observational no-go theorem.

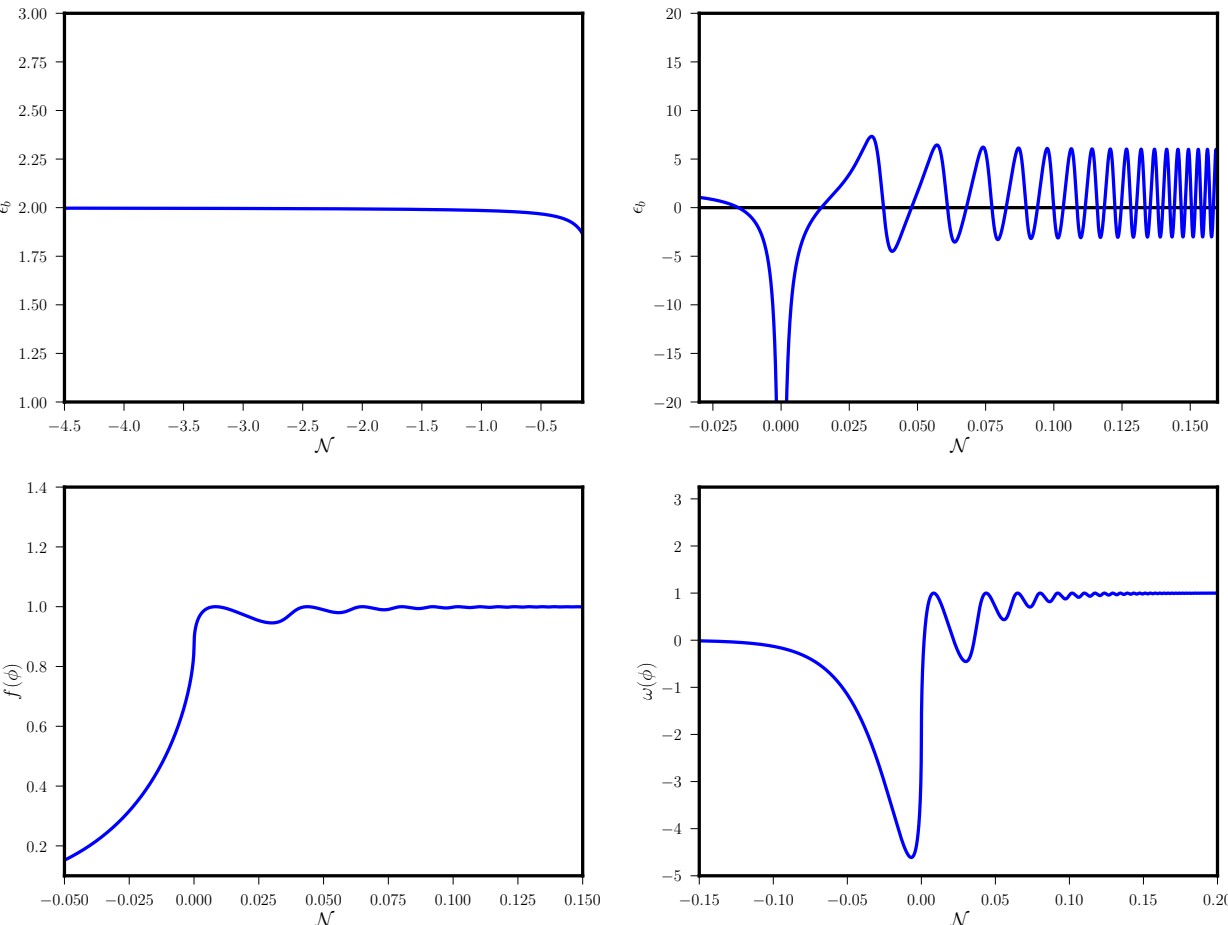

**Figure 1.** The slow-roll parameter in the contracting region (**top-left**), the slow-roll parameter in the bouncing region (**top-right**), the coupling function (**bottom-left**), and $\omega(\phi)$ (**bottom-right**) in the non-minimal theory (4) are plotted for the chaotic bouncing model with $\alpha = 1$ with respect to the e-N-fold time convention $\mathcal{N}$ ($\mathcal{N}^2 \equiv 2\ln(a/a_0)$ for contraction, $\mathcal{N}$ is negative, whereas, at bounce, it becomes zero and then during expansion, it becomes positive). It is apparent that, during the contraction period, the scale factor behaves as $a(\eta) \propto (-\eta)$, and thus, $\epsilon_b \simeq 2$. One also finds that the bouncing solution is asymmetrical. The coupling function $f(\phi)$ as well as $\omega(\phi)$ become nearly unity in the reheating epoch, which arises shortly after the bounce.

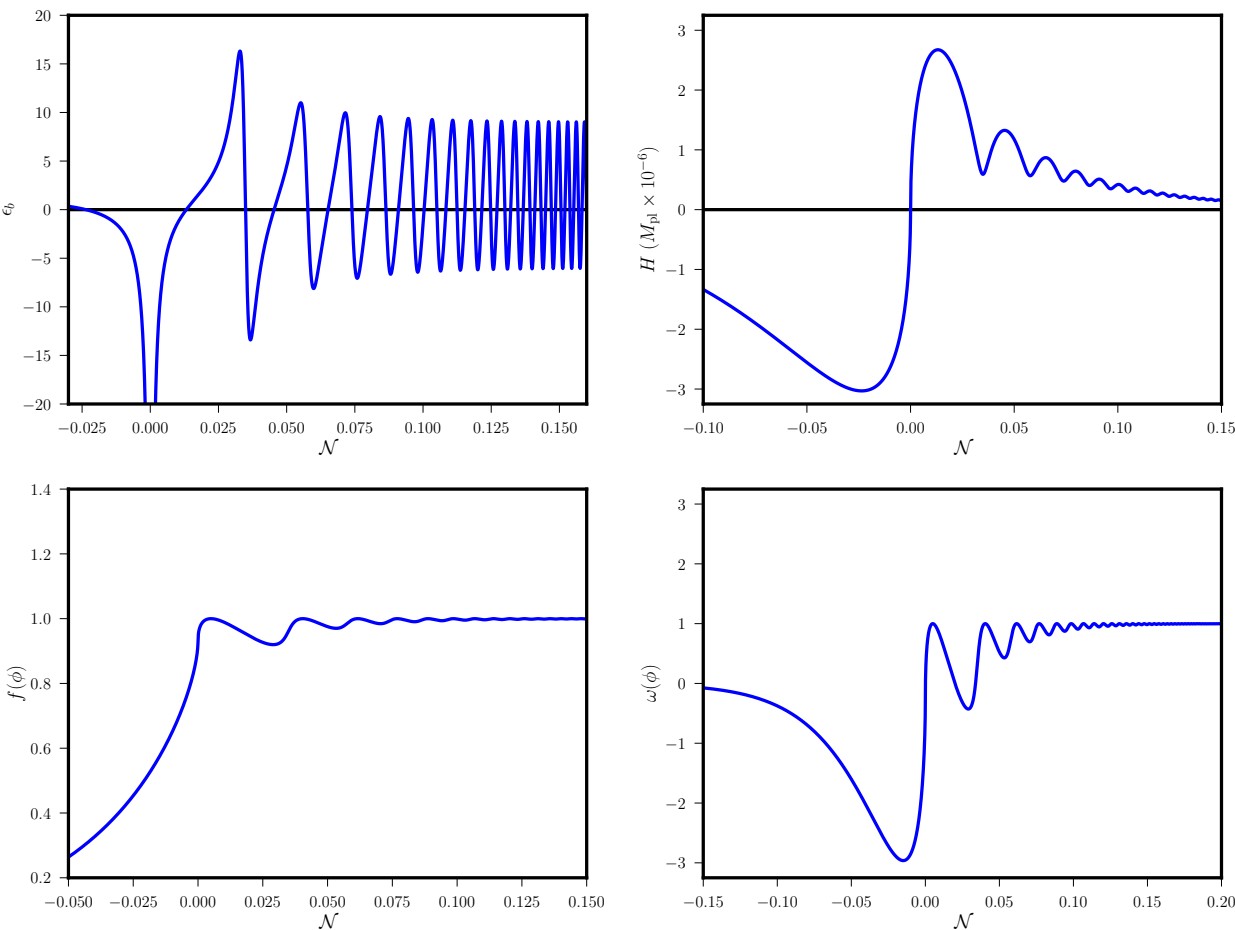

**Figure 2.** The slow-roll parameter $\epsilon_b$ (**top-left**), Hubble parameter $H_b$ (**top-right**), the coupling function $f(\phi)$ (**bottom-left**) and $\omega(\phi)$ (**bottom-right**) in the non-minimal theory (4) are plotted for the chaotic bouncing model with $\alpha = 2$ with respect to the e-N-fold time convention $\mathcal{N}$ ($\mathcal{N}^2 \equiv 2\ln(a/a_0)$, for contraction, $\mathcal{N}$ is negative, whereas, at bounce, it becomes zero and then during expansion, it becomes positive) during and after the bounce. One can see that the bouncing solution is asymmetrical. The coupling function $f(\phi)$ as well as $\omega(\phi)$ become nearly unity in the reheating epoch, which appears shortly after the bounce.

## 3. Stability Analysis

Let us now analyze the stability condition for the above system (4). Assuming the additional fluid is barotropic, in nature, with the equation of state $w_m$, the governing equations become

$$f^2\left(R_{\mu\nu} - \frac{1}{2}g_{\mu\nu}R\right) - 2\nabla_\mu f\,\nabla_\nu f - 2f\,\nabla_{\mu\nu}f + 2g_{\mu\nu}\left(\nabla^\lambda f\,\nabla_\lambda f + f\,\Box f\right)$$
$$= \frac{1}{M_{\rm Pl}^2}\left[\omega(\phi)\left(\nabla_\mu\phi\nabla_\nu\phi - \frac{1}{2}g_{\mu\nu}\nabla^\lambda\phi\nabla_\lambda\phi\right) - g_{\mu\nu}V_b + T_{\mu\nu}^{(M)}\right], \tag{10}$$

$$\Box\phi + \frac{1}{2\omega(\phi)}\left(\omega_{,\phi}\,\nabla^\lambda\phi\nabla_\lambda\phi - 2V_{,\phi} + 2M_{\rm Pl}^2\,f\,f_{,\phi}\,R\right) = 0, \tag{11}$$

where, $T_{\mu\nu}^{(M)}$ is the energy-momentum tensor of the additional barotropic fluid in the non-minimal theory (4), satisfying

$$\nabla^\mu T_{\mu\nu}^{(M)} = 0. \tag{12}$$

We have not used any subscript $b$ since, from now on, all concerning equations will correspond only to the non-minimal theory. These equations, using Friedmann–Robertson–Walker (FRW) homogeneous and isotropic Universe with the line element

$$ds^2 = -dt^2 + a^2(t)\,dx^2, \tag{13}$$

turn into

$$3f^2 H^2 = -6H f f_{,\phi} + \frac{1}{M_{\text{Pl}}^2}\left(\frac{1}{2}\omega\dot{\phi}^2 + V(\phi) + \rho_M\right) = 0, \tag{14}$$

$$\ddot{\phi} + 3H\dot{\phi} + \frac{1}{2\omega}\left(\omega_{,\phi}\dot{\phi}^2 + V_{,\phi} - 2M_{\text{Pl}}^2 f f_{,\phi} R\right) = 0, \tag{15}$$

$$\dot{\rho}_M + 3H(\rho_M + P_M) = 0, \tag{16}$$

where $\rho_M \equiv -T^{(M)0}_{\phantom{(M)}0}$ and $T^i_j \equiv P_M \delta^i_j = w_m \rho_M \delta^i_j$ are the energy density and the pressure of the barotropic fluid with $w_m$ being the equation of state. To study the stability analysis, instead of using quantities with dimensions, we can express and re-write these field equations in terms of dimensionless quantities. These are defined as

$$x \equiv \frac{\dot{\phi}}{\sqrt{6}M_{\text{Pl}}\,H}, \quad y \equiv \frac{\sqrt{V}}{\sqrt{3}M_{\text{Pl}}\,f\,H} \tag{17}$$

Using these dimensionless variables, one can quickly express the evolution equation of them as

$$\begin{aligned}
\frac{dx}{dN} \equiv \frac{1}{H}\frac{dx}{dt} = &-\frac{1}{2\gamma_e^4}(36(3w_m - 1)(1+\alpha)^4 x^3 - 18\sqrt{6}(3w_m - 1)(1+\alpha)^3\gamma_e x^2 - \\
&6(1+\alpha)^2\gamma_e^2(3 - 4x^2 - y^2 - 3w_m(3 - 2x^2 - y^2))\,x + \\
&\sqrt{6}(1 - 7x^2 - y^2 - 3w_m(1 - 3x^2 - y^2))(1+\alpha)\gamma_e^3 - \\
&3x((w_m - 1)(1 - x^2 - y^2) + 2y^2\alpha)\gamma_e^4 + \sqrt{6}y^2\gamma_e^5),
\end{aligned} \tag{18}$$

$$\begin{aligned}
\frac{dy}{dN} \equiv \frac{1}{H}\frac{dy}{dt} = &\frac{1}{2\gamma_e^4}((6(1 - 3w_m)(y^2 - 1)(1+\alpha)^2\gamma_e^2\,y + 3(1 + w_m + y^2 - w_m y^2 \\
&+2y^2\alpha)\gamma_e^4 + \sqrt{6}\gamma_e(12(3w_m - 1)(1+\alpha)^3\,x - 6w_m(1+\alpha)\gamma_e^2 + \gamma_e^4) + \\
&3(12(1+\alpha)^4 - 8(1+\alpha)^2\gamma_e^2 + \gamma_e^4 - w_m(6(1+\alpha)^2 - \gamma_e^2)^2)\,x^2)),
\end{aligned} \tag{19}$$

along with the constraint equation

$$\Omega_M \equiv \frac{\rho_M}{3M_{\text{Pl}}^2 f^2 H^2} = 1 - \frac{2\sqrt{6}(1+\alpha)}{\gamma_e}x + \left(\frac{6(1+\alpha)^2}{\gamma_e^2} - 1\right)x^2 - y^2, \quad \gamma_e \equiv M_{\text{Pl}}\frac{V_{I,\phi}}{V_I}. \tag{20}$$

$N$ is defined as the e-folding number in the non-minimal frame $\equiv \ln(a/a_0)$, $a_0$ is the initial value of the scale factor and $V_I(\phi)$ is the potential in the minimal Einstein frame and is related to $V(\phi)$ through Equation (5). Note that, the model parameter $\gamma_e$ actually represents a parameter in the Einstein frame which is completely arbitrary and can be fixed from the observations. For example, while near the pivot scale, for chaotic inflation, $\gamma_e \sim 0.01$, for Starobinsky model, at the same scale, it becomes $\sim 0.003$. However, chaotic inflation has already been ruled out from the observations while, the Starobinsky model remains consistent with it and one of the most important models in the paradigm. We should stress that the phase during the stability is analyzed is the contraction in a bouncing model and therefore, instead of the e-N-fold time convention, here we are using the e-fold time convention.

One can also express the slow-roll parameter $\epsilon_b$ in terms of these dimensionless parameters as

$$
\begin{aligned}
\epsilon_b \equiv -\frac{\dot{H}}{H^2} \;=\; & \frac{1}{2\gamma_e^4}\Big(6(1-3w_m)(1-y^2)(1+\alpha)^2\gamma_e^2 + 3(1+w_m+y^2-w_m y^2 \\
& +2y^2\alpha)\gamma_e^4 + 2\sqrt{6}\gamma_e(3w_m-1)(1+\alpha)(6(1+\alpha)^2-\gamma_e^2)\,x + \\
& 3(12(1+\alpha)^4 - 8(1+\alpha)^2\gamma_e^2 + \gamma_e^4 - w_m(6(1+\alpha)^2-\gamma_e^2)^2)\,x^2\Big). \quad (21)
\end{aligned}
$$

Using the above expression, one can define the effective equation of state as

$$
w_{\text{eff}} = -1 + \frac{2}{3}\,\epsilon_b. \tag{22}
$$

Although $\gamma_e$ is a function of $\phi$ and thus, it is dynamical, the minimal theory that we are considering leads to slow-roll inflation and thus one can consider it to be nearly constant, similar to exponential potential.

In this system, as one can see, there exist three model parameters: $\{\alpha, w_m, \gamma_e\}$. $w_m$ is arbitrary as it describes the nature of the additional field. For example, in the case of anisotropic stress fluid that corresponds to the stiff matter, $w_m = 1$. This field results in the BKL instability that, later, we will investigate. As mentioned above, $\gamma_e$ is nearly constant and hence, we are left with one model parameter $\alpha$ that can be constrained. In the next section, we will try to constrain it using the Lyapunov exponents.

### 3.1. Fixed Points and the Bouncing Solution

Using the evolution equation, one can find the fixed/critical point of the system and these points correspond to the solutions of the system, in this case, the Universe depicted by the non-minimal theory (4). These can be found by setting Equations (18) and (19) to be equal to zero, i.e., the velocities of $x$ and $y$ vanishes at these points. There are seven such fixed points:

$$
1. \quad x_1^* = \frac{\gamma_e}{\sqrt{6}\,\alpha}, \quad y_1^* = -\frac{\sqrt{6-\gamma_e^2}}{\sqrt{6}\,\alpha} \tag{23}
$$

$$
2. \quad x_2^* = \frac{\gamma_e}{\sqrt{6}\,\alpha}, \quad y_2^* = \frac{\sqrt{6-\gamma_e^2}}{\sqrt{6}\,\alpha} \tag{24}
$$

The sign of the $y^*$ signifies whether the Universe is expanding or contracting. Since, $H$ appears in the denominator in the expression of $y$ in (17), the negative sign of $y$ leads to contraction (bouncing), while positive $y$ corresponds to expanding Universe. Therefore, $\{x_1^*, y_1^*\}$ leads to contraction, and $\{x_2^*, y_2^*\}$ implies the expanding Universe. In fact, $\{x_1^*, y_1^*\}$ is our desired solution and the scalar field dominated solution as the fractional energy density of the additional fluid and the effective equation of state (see Equation (22)) of corresponds to this point are

$$
\Omega_M^{(1,2)} = 0 \tag{25}
$$

$$
w_{\text{eff}}^{(1,2)} = \frac{2-\alpha-\gamma_e^2}{3\,\alpha} \tag{26}
$$

which corresponds to the the scale factor solution as

$$
a_b(-\eta) \propto (-\eta)^\beta, \qquad \beta \equiv \frac{2\,\alpha}{2-\gamma_e^2}. \tag{27}
$$

As you can see, the exponent is not exactly $\alpha$ but $\beta$, which is contrary to the Equation (6). However, as mentioned before, $\gamma_e \ll 1$ for slow-roll models and only by using such model, we achieved such non-minimal bounce, and therefore, $\beta \simeq \alpha$. Therefore, $\{x_1^*, y_1^*\}$

corresponds to our bouncing solution that we will investigate thoroughly in the next section. Other fixed points are

$$3. \quad x_3^* = \frac{\sqrt{6}\,\gamma_e}{6(1+\alpha) - \sqrt{6}\,\gamma_e}, \quad y_3^* = 0 \tag{28}$$

$$4. \quad x_4^* = -\frac{\sqrt{6}\,\gamma_e}{6(1+\alpha) - \sqrt{6}\,\gamma_e}, \quad y_4^* = 0 \tag{29}$$

$$5. \quad x_5^* = -\frac{\sqrt{\frac{3}{2}}(1+w_m)\gamma_e}{-4(1+\alpha) + \gamma_e^2},$$

$$y_5^* = -\frac{\sqrt{2(1-3w_m)^2\alpha(1+\alpha)^2 + \alpha(1-3w_m^2 - 2\alpha + 6w_m(1+\alpha))\gamma_e^2}}{(\sqrt{2\alpha}(4(1+\alpha) - \gamma_e^2))} \tag{30}$$

$$6. \quad x_6^* = -\frac{\sqrt{\frac{3}{2}}(1+w_m)\gamma_e}{-4(1+\alpha) + \gamma_e^2},$$

$$y_6^* = \frac{\sqrt{2(1-3w_m)^2\alpha(1+\alpha)^2 + \alpha(1-3w_m^2 - 2\alpha + 6w_m(1+\alpha))\gamma_e^2}}{(\sqrt{2\alpha}(4(1+\alpha) - \gamma_e^2))} \tag{31}$$

$$7. \quad x_7^* = \frac{\sqrt{\frac{2}{3}}(3w_m - 1)(1+\alpha)\gamma_e}{2(-1+3w_m)(1+\alpha)^2 - (-1+w_m)\gamma_e^2}, \quad y_7^* = 0 \tag{32}$$

Since our desired fixed point is $\{x_1^*, y_1^*\}$, in this work, we will focus only on this point. In the next subsection, we will explore the Lyapunov exponents that determine the stability of the solution, and then later, we will focus on the behavior of the Universe close to the fixed point.

*3.2. Lyapunov Exponent and the Stability of the Bouncing Solution*

Now, in order to study the stability of these fixed points, we need to linearize the Equations (18) and (19) as

$$\begin{pmatrix} \dfrac{\mathrm{d}\delta x}{\mathrm{d}N} \\ \dfrac{\mathrm{d}\delta y}{\mathrm{d}N} \end{pmatrix} = \begin{pmatrix} \dfrac{\partial A(x,y)}{\partial x}\Big|_* & \dfrac{\partial A(x,y)}{\partial y}\Big|_* \\ \dfrac{\partial B(x,y)}{\partial x}\Big|_* & \dfrac{\partial B(x,y)}{\partial y}\Big|_* \end{pmatrix} \begin{pmatrix} \delta x \\ \delta y \end{pmatrix}, \tag{33}$$

where, $A(x,y)$ and $B(x,y)$ are the right-hand side of (18) and (19), respectively. $|_*$ denotes the value at the fixed point. $\delta x$ and $\delta y$ are the deviations of the fixed points. By linearizing equations, we assume that we are studying the stability condition in the vicinity of the fixed points, i.e., the deviation from the fixed points are very small. We are considering homogeneous and isotropic background and assume that the anisotropies and inhomogeneities caused by the deviations do not impact on the background. This can be achieved by assuming that the energy fractions due to the deviations are extremely small.

In order to check the stability of the fixed points, we need to evaluate the eigenvalues of this matrix. These eigenvalues are the Lyapunov exponents that arise in the evolution of both $\delta x$ and $\delta y$ and thus, only these values determine the stability of the fixed points as

$$\delta x = C_{11}\,e^{\lambda_1 N} + C_{12}\,e^{\lambda_2 N}$$
$$\delta y = C_{21}\,e^{\lambda_1 N} + C_{22}\,e^{\lambda_2 N}$$

As defined earlier, $N$ is the e-folding number; in an expanding Universe, while it increases, it decreases for contracting solution, and therefore, if both the eigenvalues are negative (positive) in an expanding (contracting) Universe, then $\delta x$ and $\delta y$ approach zero as $N$ approaches $\infty$ $(-\infty)$. This implies that the deviation from the actual trajectory

reduces with time and the new trajectory returns to the original trajectory asymptotically in time. In other words, both eigenvalues being negative (positive) in an expanding (contracting) Universe implies that the fixed point is stable and the corresponding solution is an attractor solution.

Let us now calculate the Lyapunov exponents for our desired fixed point (23). It takes the form

$$\lambda_1^{(1)} = \frac{4 + \alpha - 3w_m \alpha - \gamma_e^2}{\alpha}, \quad \lambda_2^{(1)} = \frac{6 - \gamma_e^2}{2\alpha} \tag{34}$$

The above expression bears one of the main results of this work. As one can see properly since the desired fixed point is the bouncing solution, we require both of the $\lambda$'s to be positive. Few things to note: since $\alpha = -1$ represents the minimal solution as mentioned before (coupling function (7) becomes unity), one can easily obtain the results in Refs. [16,17] by substituting $\alpha = -1$. Since $\alpha$, $w_m$ and $\gamma_e$ are positive, increasing values of $w_m$ and $\gamma_e$ narrow the permissible regime of $\alpha$ for being an attractor. Anisotropic fluid or the stress fluid resembles similar to a stiff matter with the equation of state $w_m = 1$ and the energy density $\rho_M \sim a^{-6}$, where $a$ is the scale factor. Therefore, since BKL instability is caused by it, the condition to evade the BKL instability can easily be obtained by substituting $w_m = 1$:

$$\gamma_e^2 < 6, \qquad \alpha < 2 - \frac{\gamma_e^2}{2}. \tag{35}$$

Therefore, for exponential potential, $\gamma_e$ is constant and positive domain of $\alpha$ can only be obtained if $\gamma_e < 2$. In the case of slow-roll, $\gamma_e \ll 1$ and therefore, for slow-roll inflationary models which are conformal to the non-minimal bounce, the condition becomes

$$0 < \alpha < 2. \tag{36}$$

as the bouncing solution can only be obtained for $\alpha > 0$. This is the main result of this article as, now, $\alpha$ cannot possess arbitrary positive values for a stable viable solution. In the next section, in order to understand how the deviations impact the system near the fixed point, we will study the phase space in the concerned region.

## 4. Evolution of the Deviations

Using the eigenvalues and the eigenvectors, one can easily solve (33). For the fixed point (23), it becomes

$$\delta x^{(1)}(N) = A_{11}(\alpha, w_m, \gamma_e) \, e^{\lambda_1^{(1)} N} + A_{12}(\alpha, w_m, \gamma_e) \, e^{\lambda_2^{(1)} N} \tag{37}$$

$$\delta y^{(1)}(N) = A_{21}(\alpha, w_m, \gamma_e) \, e^{\lambda_1^{(1)} N} + A_{22}(\alpha, w_m, \gamma_e) \, e^{\lambda_2^{(1)} N}, \tag{38}$$

where, $\lambda_1^{(1)}$ and $\lambda_2^{(1)}$ are given in (34). The $A$'s can be evaluated as

$$A_{11} = \frac{(2(3w_m - 1)(1 + \alpha) - (w_m - 1)\gamma_e^2)\left((6 + 6\alpha - \gamma_e^2)\,\delta x_0 + \gamma_e\sqrt{6 - \gamma_e^2}\,\delta y_0\right)}{\alpha\gamma_e^2(\gamma_e^2 - 2 - 2(1 - 3w_m)\alpha)} \tag{39}$$

$$A_{12} = \frac{1}{\alpha\gamma_e^2(\gamma_e^2 - 2 - 2(1 - 3w_m)\alpha)}\left((\gamma_e^2 - 6)(2(3w_m - 1)(1 + \alpha)^2 + \right.$$
$$\left. (1 - w_m + \alpha)\gamma_e^2)\,\delta x_0 + \gamma_e\sqrt{6 - \gamma_e^2}(2(1 - 3w_m)(1 + \alpha) + (w_m - 1)\gamma_e^2)\,\delta y_0\right) \tag{40}$$

$$A_{21} = \frac{1}{\alpha\gamma_e^3(\gamma_e^2 - 2 - 2(1 - 3w_m)\alpha)}\left((2(3w_m - 1)(1 + \alpha)^2 + \right.$$
$$\left. (1 - w_m + \alpha)\gamma_e^2)(\sqrt{6 - \gamma_e^2}(\gamma_e^2 - 6 - 6\alpha)\delta x_0 + \gamma_e(\gamma_e^2 - 6)\delta y_0)\right) \tag{41}$$

$$A_{22} = \frac{1}{\alpha\gamma_e^3(\gamma_e^2 - 2 - 2(1 - 3w_m)\alpha)}\left((6 + 6\alpha - \gamma_e^2)(\sqrt{6 - \gamma_e^2}(2(3w_m - 1) \right.$$
$$\left. (1 + \alpha)^2 + (1 - w_m + \alpha)\gamma_e^2)\,\delta x_0 + \gamma_e(2(3w_m - 1)(1 + \alpha) - (w_m - 1)\gamma_e^2)\,\delta y_0)\right) \tag{42}$$

These solutions are obtained by using the initial conditions $\delta x(0) = \delta x_0$, $\delta y(0) = \delta y_0$. The slow-roll parameter can also be obtained in the vicinity of the fixed point (23) by perturbing the Equation (21) and this becomes

$$\epsilon_b = \epsilon_b^{(1)} + \epsilon_{bx}\,\delta x(N) + \epsilon_{by}\,\delta y(N), \quad \epsilon_b^{(1)} = 1 + \frac{1}{\alpha} - \frac{\gamma_e^2}{2\alpha}, \tag{43}$$

where,

$$\epsilon_{bx}^{(1)} = \frac{\sqrt{\frac{3}{2}}(6 + 12\alpha + 6\alpha^2 - \gamma_e^2)(2 + 2\alpha - \gamma_e^2 - w_m(6 + 6\alpha - \gamma_e^2))}{\alpha\gamma_e^3} \tag{44}$$

$$\epsilon_{by}^{(1)} = \frac{\sqrt{9 - \frac{3\gamma_e^2}{2}}(2 + 2\alpha^2 - \gamma_e^2 + 2\alpha(2 - \gamma_e^2) - w_m(6 + 12\alpha + 6\alpha^2 - \gamma_e^2))}{\alpha\gamma_e^2}. \tag{45}$$

Once we obtain the solution expression for the slow-roll parameter, we can solve the energy density equation as $\epsilon(N) = -\mathrm{d}_N \ln H(N)$ and by performing the integral, one can easily solve for $H$. It becomes, $H^2(N) = H_0^2\exp(-2\int \mathrm{d}N\,\epsilon(N))$, $H_0$ is the initial value of the Hubble parameter. Using (37) and (43), and by using the fact that, $\delta x_0$ and $\delta y_0$ are tiny and act like perturbations, we obtain the evolution of the Hubble parameter as

$$\frac{H^2}{H_0^2} = (1 - \Omega_{m1} - \Omega_{m2})\left(\frac{a}{a_0}\right)^{-2\epsilon_b^{(1)}} + \Omega_{m1}\left(\frac{a}{a_0}\right)^{\lambda_1^{(1)} - 2\epsilon_b^{(1)}} + \Omega_{m2}\left(\frac{a}{a_0}\right)^{\lambda_2^{(1)} - 2\epsilon_b^{(1)}} \tag{46}$$

where,

$$\Omega_{m1} = -2\frac{\epsilon_{bx}^{(1)} A_{11}}{\lambda_1^{(1)}} - 2\frac{\epsilon_{by}^{(1)} A_{21}}{\lambda_1^{(1)}} \tag{47}$$

$$\Omega_{m2} = -2\frac{\epsilon_{bx}^{(1)} A_{12}}{\lambda_2^{(1)}} - 2\frac{\epsilon_{by}^{(1)} A_{22}}{\lambda_2^{(1)}} \tag{48}$$

One can easily obtain the the above expressions in terms of model parameters $\{\alpha, w_m, \gamma_e\}$ by using relations (39)–(42), (44) and (45). It is clear from the above equation that the system behaves as the Universe contains three fluids, where the leading solution corresponds the scalar field dominated solution, in the case of additional fluids having the initial energy density fraction $\Omega_{m1}$ and $\Omega_{m2}$, the energy density of them grow/decay as $\rho_{m1} \propto \left(\frac{a}{a_0}\right)^{\lambda_1^{(1)} - 2\epsilon_b^{(1)}}$ and $\rho_{m2} \propto \left(\frac{a}{a_0}\right)^{\lambda_2^{(1)} - 2\epsilon_b^{(1)}}$. It is apparent that, relative to

the leading order term with $\left(\frac{a}{a_0}\right)^{-2\epsilon_b^{(1)}}$, the other two decay/grow with the factor $\left(\frac{a}{a_0}\right)^{\lambda_1^{(1)}}$ and $\left(\frac{a}{a_0}\right)^{\lambda_2^{(1)}}$, respectively. Therefore, for a contracting solution, $\lambda$'s being positive signifies that the those additional fluids represented by the $\Omega_{m1}$ and $\Omega_{m2}$ decay. This can even be seen from the expression of the energy fraction of the additional fluid (20). This becomes

$$\Omega_M = \frac{\sqrt{\frac{2}{3}}\left((6 + 6\alpha - \gamma_e^2)\,\delta x_0 + \gamma_e\sqrt{6 - \gamma_e^2}\,\delta y_0\right)}{(\alpha\gamma_e)}\left(\frac{a}{a_0}\right)^{\lambda_1\,N}. \tag{49}$$

From the above expression, it becomes obvious that, for a bouncing solution, if $\lambda$'s are positive, the energy fraction vanishes after a while. To understand the correct nature of it, below, we presented three different scenarios with different choices of model parameters that try to help us realize the system.

### 4.1. Radiation Bounce

As we know from (36), $\alpha$ must be positive and less than 2 in order to satisfy bouncing solution as well as to evade the BKL instability. Therefore, in the first example, we consider radiation bounce where $\alpha = 1$, i.e., the scale factor solution approximately becomes $a_b(\eta) \propto (-\eta)$. We consider the additional fluid as the anisotropic stress fluid with stiff equation of state $w_m = 1$. Assuming typical conformal slow-roll model (e.g., chaotic inflation with $V_I(\phi) = \frac{1}{2}\,m^2\,\phi^2$) with $\gamma_e \approx 0.01$, the Hubble solution takes the form

$$\frac{H^2}{H_0^2} = (1 - \Omega_{m1} - \Omega_{m2})\left(\frac{a}{a_0}\right)^{-4} + \Omega_{m1}\left(\frac{a}{a_0}\right)^{-2} + \Omega_{m1}\left(\frac{a}{a_0}\right)^{-1} \tag{50}$$

with

$$\frac{\Omega_{m1}}{\Omega_{m2}} = \frac{-1.125\,\delta x_0 + 0.002\,\delta y_0}{\delta x_0 + 0.002\,\delta y_0}. \tag{51}$$

The equation represents the attractor nature of the solution as the leading order solution, i.e., the effective radiation matter solution dominates over the additional fluid evolution. The actual energy density fraction of the additional fluid (49) takes the form

$$\Omega_M = (980\,\delta x_0 + 2\,\delta y_0)\left(\frac{a}{a_0}\right)^2. \tag{52}$$

For bouncing solution, as it is obvious, it quickly vanishes after a while.

### 4.2. Comparing with the Minimal Scenario

As it has been mentioned before, negative $\alpha$ represents the expanding solution, and $\alpha = -1$ reduces to the minimal Einstein inflationary scenario as for this value, the coupling function becomes unity. In this subsection, we will compare the above result with the minimal inflationary case. Again using the similar value of the additional fluid equation of state as well the $\gamma_e$, i.e., $w_m = 1$, $\gamma_e \approx 0.01$, the Hubble solution becomes

$$\frac{H^2}{H_0^2} = (1 - \Omega_{m1} - \Omega_{m2}) + \Omega_{m1}\left(\frac{a}{a_0}\right)^{-6} + \Omega_{m1}\left(\frac{a}{a_0}\right)^{-3}, \tag{53}$$

with

$$\frac{\Omega_{m1}}{\Omega_{m2}} = -0.05 + 122.47\,\frac{\delta y_0}{\delta x_0}. \tag{54}$$

Notice that, in this case, the additional fluids with $\Omega_{m1}$ and $\Omega_{m2}$ initial mass fractions decay faster compared to the radiation bounce scenario as the Lyapunov exponents, i.e., the

$\lambda$'s are having high values. This can even be seen from the expression of energy fraction of the additional fluid (49):

$$\Omega_M = (0.008\, \delta x_0 - 2\, \delta y_0) \left( \frac{a}{a_0} \right)^{-6} \tag{55}$$

It vanishes faster than that of the radiation bounce case. Therefore, one may consider, by judging the capability of stability solution, minimal inflationary models are preferred compared to the non-minimal bouncing models. In the next subsection, we will show that, this is clearly not the case.

*4.3. Comparing with the Viable Ekpyrosis*

In Ref. [34], it has been studied extensively and showed that the ekpyrosis is a super-attractor (although it has been studied for minimal models, one can easily extend it for non-minimal solution as well. See Ref. [56] in this context). It can also been seen from (34) as $\alpha$ appears in the denominator in the expressions. This implies that the smaller values of $\alpha$ leads to higher values of Lyapunov exponents and this, in return, implies the solutions are, in comparison, more stable. Consider the example of $\alpha = 0.1$: the solution leads to the ekpyrosis solution. By using similar values of $w_m$ and $\gamma_e$, one can obtain the Hubble solution for the ekpyrosis as

$$\frac{H^2}{H_0^2} = (1 - \Omega_{m1} - \Omega_{m2}) \left( \frac{a}{a_0} \right)^{-22} + \Omega_{m1} \left( \frac{a}{a_0} \right)^{16} + \Omega_{m1} \left( \frac{a}{a_0} \right)^{8} \tag{56}$$

with

$$\frac{\Omega_{m1}}{\Omega_{m2}} = \frac{-0.95\, \delta x_0 - 0.003\, \delta y_0}{\delta x_0 + 0.003\, \delta y_0}. \tag{57}$$

Again, compared to the leading solution, the externals fluids decay much faster than the minimal case, shown above. This is because the Lyapunov exponents are, in this case, 38 and 30, respectively. For instance, the energy fraction of the additional fluid becomes

$$\Omega_M = (5389\, \delta x_0 + 20\, \delta y_0) \left( \frac{a}{a_0} \right)^{38}. \tag{58}$$

Therefore, even if all the examples, that are presented in this work, are in line with the observations, there is a difference in the behavior of the stability and among them, ekpyrosis is the most stable, and in this sense, the ekpyrosis is the most preferred, which again, has been re-established.

**5. Conclusions**

In this article, we construct a non-minimal bouncing model that is conformal to the scalar sector of the minimal inflationary model. However, along with the presence of the additional matter in the system, the minimal and non-minimal models do not conform and therefore, the stability of the two systems is not related by the conformal factor. $\alpha$ is the main model parameter and by setting it to be equal to $-1$, one can arrive at the stability conditions of the minimal model as well as it had been studied before in the literature. While, the scalar part of the non-minimal bounce is conformal to that of the minimal inflationary model, choosing a viable inflationary model that satisfies the observational constraints, the non-minimal bouncing model also satisfies the observation constraints with the inherent assumption that the model solution is an attractor. If the solution is not stable, then even the quantum fluctuations can grow and diverge and the system becomes unstable. In this article, therefore, we study the stability of the non-minimal bouncing model with the presence of an additional fluid with the equation of state $w_m$. We show that, while $\alpha$ being positive leads to a viable bouncing solution, not all positive values of it

leads to a stable solution, and to avoid BKL instability, we obtained the constraint in $\alpha$ as the domain for stability as well as bounce as $0 < \alpha < 2$. We also showed that the ekpyrosis is the best attractor among them and thus is always preferred.

While we studied the system for homogeneous and isotropic background, one can easily extend the work for homogeneous and anisotropic systems as well and may arrive at a similar condition. One can even go beyond the homogeneous system and study the stability using numerical relativity. However, this is beyond the scope of this article. We did not consider the negative values of $\alpha$ as well as $\alpha = 0$ case. The last one signifies the emergent gravity scenario. Apart from that, we also did not study the behavior of the other fixed points (28)–(32). While in the minimal scenario, most of them are unstable, in our case, it may not be the same and may impact the system heavily. We reserve these works for our future prospects.

**Funding:** This research received no external funding.

**Institutional Review Board Statement:** Not applicable.

**Informed Consent Statement:** Not applicable.

**Data Availability Statement:** Not applicable.

**Acknowledgments:** The author thanks the Indian Institute of Technology Madras (IIT Madras), Chennai, India for support through the institute postdoctoral fellowship.

**Conflicts of Interest:** The author declares no conflict of interest.

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
