# Peer review of "Stability of a Viable Non-Minimal Bounce"

_universe, doi:10.3390/universe7030062_

Round 1

Reviewer 1 Report

The paper explores the possibility of a “bounce” model that manages to circumvent BKL instability and is compatible with the observational restrictions. The proposed model is based on a scalar with non-minimal coupling (after conformal transformation from Einstein to Jordan  frame) and a sector of matter with an external fluid. After presenting the evolution equations of the system, the author determines the fixed points that correspond to the bounce solution and, when evaluating the stability of these points, he verifies that in order to guarantee the bounce solution and avoid BKL instability, a restrictive condition for model parameters. Besides, the work also studies the phase space in the region of the fixed points, and analyzes three scenarios with different values ​​for the model parameters, within the conditions of stability.

Let me say that I have very restricted experience with the bouncemodels. However, at least for me, the preprint looks correct and is sufficiently well-written to be published in Universe. However, it would be for good to take into account the details listed below. 

Some typos:

1) in line 56 (page 2) it says “… evading the stability…”. This passage seems to me to be incorrect: I think it should replace stability with instability;

2) on line 85 (page 2), there is an excess of that;

3) the second equation (not listed) after line 270 (page 9) corresponds to the deviation \delta y. Therefore, \delta x 
must be replaced by \delta y in the first member of this equation;

4) line 285 (page 10) contains the word “Since” with the first letter in uppercase in the middle of the sentence;

5) equations (40), (41) and (42), on p. 11, are messed up. I believe that it is possible to better organize the denominator 
of these expressions;

6) on p. 12 there is no equation listed as (49).

Comments / Suggestions:

7) The solution of the scale factor of the model is not exact, thus making the numerical analysis of the bounce structure limited to the results presented in figure 1 (including, these are for a scenario with \alpha = 2, which does not satisfy the condition of stability). So, it might be interesting if the author added other plots that include the domain 0 < \alpha <2;

8) In my opinion, the author could better explain how the model manages to satisfy the observational restrictions.

Reviewer 2 Report

The paper reports a detailed study on the possibility of constructing a viable model of bouncing Universe. The computations are correct, and the obtained results may have interesting cosmological applications. 

As a possible small improvement, the author may wish to note (maybe in the Introduction) that the problem of evading the classical no-go theorems, and of obtaining non-singular bouncing solutions, has been addressed in the past also in the context of the so-called pre-big bang scenario based on the string cosmology equations, as reported for instance in a recent review on this subject:
M.~Gasperini and G.~Veneziano,
``String Theory and Pre-big bang Cosmology,''
Nuovo Cim. C \textbf{38} (2016) no.5, 160,
doi:10.1393/ncc/i2015-15160-8,
[arXiv:hep-th/0703055 [hep-th]].

By the way, the conformal transformation (3), defining the bouncing metric of this paper, leads to a non-minimal action very similar, in many respects, to the gravi-dilaton string effective action, with the inclusion of those non-perturbative string-coupling corrections  which are indeed a crucial ingredient (as stressed in the above quoted paper) for implementing a successfull bouncing scenario. 

After this small improvement the paper, in my opinion, can be accepted for publication in the journal "Universe". 
